# Eicosapentaenoic Acid Modulates Transient Receptor Potential V1 Expression in Specific Brain Areas in a Mouse Fibromyalgia Pain Model

**DOI:** 10.3390/ijms25052901

**Published:** 2024-03-01

**Authors:** Hsien-Yin Liao, Chia-Ming Yen, I-Han Hsiao, Hsin-Cheng Hsu, Yi-Wen Lin

**Affiliations:** 1College of Chinese Medicine, School of Post-Baccalaureate Chinese Medicine, China Medical University, Taichung 40402, Taiwan; 017215@tool.caaumed.org.tw; 2Department of Anesthesiology, Taichung Tzu Chi Hospital, Buddhist Tzu Chi Medical Foundation, Taichung 42743, Taiwan; terry1974@mail.cmu.edu.tw; 3School of Post-Baccalaureate Chinese Medicine, Tzu Chi University, Hualien 97004, Taiwan; 4College of Chinese Medicine, School of Medicine, China Medical University, Taichung 404328, Taiwan; 018309@tool.caaumed.org.tw; 5Department of Neurosurgery, China Medical University Hospital, Taichung 404332, Taiwan; 6Department of Traditional Chinese Medicine, China Medical University Hsinchu Hospital, Hsinchu 302, Taiwan; 7College of Chinese Medicine, Graduate Institute of Acupuncture Science, China Medical University, 91 Hsueh-Shih Road, Taichung 40402, Taiwan; 8Chinese Medicine Research Center, China Medical University, Taichung 40402, Taiwan

**Keywords:** eicosapentaenoic acid, fibromyalgia, thalamus, mPFC, TRPV1

## Abstract

Pain is an unpleasant sensory and emotional experience accompanied by tissue injury. Often, an individual’s experience can be influenced by different physiological, psychological, and social factors. Fibromyalgia, one of the most difficult-to-treat types of pain, is characterized by general muscle pain accompanied by obesity, fatigue, sleep, and memory and psychological concerns. Fibromyalgia increases nociceptive sensations via central sensitization in the brain and spinal cord level. We used intermittent cold stress to create a mouse fibromyalgia pain model via a von Frey test (day 0: 3.69 ± 0.14 g; day 5: 2.13 ± 0.12 g). Mechanical pain could be reversed by eicosapentaenoic acid (EPA) administration (day 0: 3.72 ± 0.14 g; day 5: 3.69 ± 0.13 g). A similar trend could also be observed for thermal hyperalgesia. The levels of elements in the transient receptor potential V1 (TRPV1) signaling pathway were increased in the ascending pain pathway, including the thalamus, medial prefrontal cortex, somatosensory cortex, anterior cingulate cortex, and cerebellum. EPA intake significantly attenuated this overexpression. A novel chemogenetics method was used to inhibit SSC and ACC activities, which presented an analgesic effect through the TRPV1 downstream pathway. The present results provide insights into the role of the TRPV1 signaling pathway for fibromyalgia and its potential as a clinical target.

## 1. Introduction

Fibromyalgia is often accompanied by widespread musculoskeletal pain, resulting in fatigue, sleep disturbance, memory loss, and psychiatric problems. Classic fibromyalgia symptoms often begin after a triggering event and progressively accumulate without eliciting occurrence. Fibromyalgia often occurs in women who also have tension headaches, irritable bowel syndrome, anxiety, and depression [1,2,3]. Currently, there is no cure for fibromyalgia, medication can only manage the symptoms. Nutrients, exercise, meditation, yoga, and relaxation may help [4,5,6]. Fibromyalgia can result from repeated nerve stimulation, also called central sensitization, or changes in the overexpression of neurotransmitters or neuromodulators in the brain. These phenomena can result from genetic mutations, infections, and psychological stress. Fibromyalgia affects about 1–8% of the population [7,8,9]. The American College of Rheumatology reorganized the diagnostic principles for fibromyalgia. They used the widespread pain index (WPI) and symptom severity scale (SS) to evaluate fibromyalgia. The WPI assesses 19 common pain points experienced over two weeks, and the SS estimates the degrees of fatigue, waking, and cognitive symptoms. Fibromyalgia pain is defined by a WPI ≥ 7 and SS ≥ 5 or WPI 3–6 and SS ≥ 9 over three months. Duloxetine (Cymbalta), milnacipran (Savella), and pregabalin (Lyrica) are the current FDA-approved treatments for fibromyalgia. Gabapentinoids, sedatives, selective serotonin reuptake inhibitors, serotonin norepinephrine reuptake inhibitors, and tricyclic compounds are also utilized for fibromyalgia, though having several side effects. The economic impact precise cost drivers of healthcare spending for fibromyalgia symptoms are high [10,11,12].

TRPV1, which is one of the six members of the TRPV subgroup, is the first component identified and the most frequently considered with regard to pain research. Therefore, it is very important to study the mechanisms of TRPV1 in fibromyalgia pain, especially to explore its molecular mechanisms in the brain. Transient receptor potential vanilloid 1 (TRPV1) channels act as detectors of several microenvironmental signals, such as mechanical, thermal, pain, vision, or stress, and they can be widely stimulated by several related inputs. These ion channels, permeable to Na^+^ and Ca^2+^, are found in several tissues typically on the cell membrane. Accordingly, TRPV1 was selected for drug development and suggested for curing many diseases [13,14,15]. TRPV1 modulation has been clinically investigated for different indications, especially for pain treatment, as these channels are the main detectors and transducers of painful signals from the peripheral to the central nervous system. Numerous potent TRPV1 agonists and antagonists have been used in clinical trials to cure visceral, inflammatory, and neuropathic pain [16,17,18]. However, the therapeutic effect was unsatisfactory, causing several side effects. TRPV1 is considered a critical brain inflammatory sensor and pain biomarker in mouse brains [19]. Triggering TRPV1 can increase the level of mitogen-activated protein kinase (MAPK), which is involved in pain development. MAPK has three major subfamilies: phosphorylated extracellular signal-regulated protein kinase (pERK), phosphorylated p38 protein kinase (pp38), and phosphorylated c-Jun N-terminal kinase/stress-activated protein kinase (pJNK) [20]. TRPV1 downstream of the PI3K-Akt-mTOR axis was also indicated to be involved in the modulation of pain sensation [21].

Eicosapentaenoic acid (EPA) is a long-chain omega-3 polyunsaturated fatty acid mostly found in cold-water fish. EPA has beneficial effects on heart disease, high triglycerides, hypertension, and inflammation. EPA is an essential nutrient with anti-inflammatory [22], antidepression [23], neuroprotective [24], and cardioprotective effects [25]. Even though EPA is used as a dietary supplement or nutrient-based medicine, its detailed mechanisms are still unclear. EPA was also reported to have anti-inflammation and immune functions in the brain. In addition, docosahexaenoic acid (DHA) is abundant in the brain and modulates neurotransmission, neuroplasticity, and neuroprotection responses. A recent article indicated lower levels of EPA in patients with depression suggesting that EPA plays a crucial role in the pathogenesis of depression. The findings provide further support for using EPA as an alternative therapy for depression [26,27,28]. Thus, EPA has potential as an alternative therapy for depression without creating side effects [29,30,31,32,33]. Furthermore, EPA has been shown to relieve chronic pain and depression in mouse models. EPA can reduce chronic pain and depression through the regulation of interleukins [21].

In the current study, we aimed to investigate the role of TRPV1, which is the main detector and transducer of nociceptive signals, in a mice model of fibromyalgia pain. Since the precise mechanisms behind the effects and mechanisms of EPA in this model remain unknown, we evaluated EPA in the expression of the TRPV1 signaling pathway in a murine fibromyalgia pain model. We found the augmented expression and localization of TRPV1 and associated kinases in a mouse model of fibromyalgia. These increased levels of signaling mediators were found in the mouse thalamus (THA), somatosensory cortex (SSC), anterior cingulate cortex (ACC), medial prefrontal cortex (mPFC), and cerebellum. Furthermore, on the contrary, cannabinoid receptor 1 (CB1) had a different decreasing trend. These effects were reversed by EPA administration. Thus, our results demonstrate the positive effects of EPA on a mouse fibromyalgia pain model and serve to advise the clinical use of EPA.

## 2. Results

### 2.1. EPA Attenuated the Intermittent Cold Stress-Induced Fibromyalgia Pain in Mice

To address if the EPA had analgesic effects on mouse fibromyalgia pain, we performed the von Frey and Hargraves tests to detect mechanical and thermal hyperalgesia. After inducing the mice fibromyalgia model for three days, their mechanical or thermal hyperalgesia was measured. The mechanical thresholds in the fibromyalgia group, which underwent cold stress inductions, were significantly lower than those in the normal group (Figure 1A, red column, day 5: 2.13 ± 0.12 g, * *p* < 0.05, *n* = 9). EPA mitigated the mechanical hypersensitivity by increasing the withdrawal threshold measured by the von Frey test (Figure 1A, blue column, day 5: 3.69 ± 0.13 g, ^#^*p* < 0.05, *n* = 9). In the Hargreaves test, cold stress significantly decreased the thermal latency compared to the normal group, suggesting the successful induction of fibromyalgia pain. In contrast, EPA administration alleviated thermal hyperalgesia by increasing the thermal latency (Figure 1B, * *p* < 0.05, *n* = 9).

### 2.2. EPA Relieved Fibromyalgia Pain through the Regulation of the TRPV1 Signaling Pathway in the Mouse Thalamus

Central sensitization is involved in hyperalgesia from nociceptor signals in the peripheral region. We used Western blot to detect TRPV1 expression in the mouse thalamus, a crucial brain area for ascending pain processing. The induction of fibromyalgia pain via cold stress reliably increases TRPV1 expression in the thalamus (Figure 2A, black columns, * *p* < 0.05, *n* = 6), while EPA suppresses this effect after three intakes (Figure 2A, black columns, # *p* < 0.05, *n* = 6). We next considered the expression of downstream protein kinases involved in producing pain signals, such as phosphorylated PKA, PI3K, and PKC. Fibromyalgia pain significantly increases the expressions of those kinases in comparison to the normal group (Figure 2A, * *p* < 0.05, *n* = 6). This increase decreases with EPA administration (Figure 2A, # *p* < 0.05, *n* = 6). In addition, mice with fibromyalgia pain showed dramatic increases in pAkt and the pmTOR axis. Furthermore, pERK, pp38, and pJNK protein levels increase in the thalamus of fibromyalgia mice (Figure 2A, * *p* < 0.05, *n* = 6); such an upregulation is reduced by EPA administration (Figure 2A, # *p* < 0.05, *n* = 6). After the impression of protein kinase activation, the transcription factors, including pCREB and pNF-κB, initiated gene transcription. pCREB and pNF-κB expressions increased after fibromyalgia pain induction in the thalamus. EPA intake significantly decreased this upregulation. Using immunofluorescent staining, we next determined the lower expression and colocalization of TRPV1 and pERK in the normal mouse thalamus than in that of mice with fibromyalgia. EPA treatment inhibited this increase (Figure 2C; green, red, or yellow colors; *n* = 3).

### 2.3. EPA Intake Mitigated Cold Stress-Induced Fibromyalgia Pain in the Mouse Somatosensory Cortex

We next explored the effects of EPA and the TRPV1 pain-signaling pathway on the SSC. TRPV1 expression was potentiated in the mouse SSC after fibromyalgia pain induction (Figure 3A, * *p* < 0.05, *n* = 6). This overexpression was remarkably alleviated by EPA treatment (Figure 3A, # *p* < 0.05, *n* = 6). The mouse fibromyalgia model resulted in pPKA, pPI3K, and pPKC overexpressions in the SSC; this overexpression decreased with EPA treatment. Similar results were observed for pAkt, pmTOR downstream molecules in the SSC. Furthermore, cold stress increased pERK, pP38, and pJNK levels as well as those of pCREB and pNF-kB in the SSC of the fibromyalgia group (Figure 3B, * *p* < 0.05, *n* = 6), an effect reversed by EPA treatment (Figure 3B, # *p* < 0.05, *n* = 6). We next determined the lower colocalizations of TRPV1 and pERK in the normal mouse SSC than in the fibromyalgia group. This increase was attenuated by EPA treatment (Figure 3C; green, red, or yellow colors; *n* = 3).

### 2.4. EPA Treatment Attenuated Fibromyalgia Pain through TRPV1 Modulation in the ACC

After the final behavioral tests on day 5, we collected the mouse ACC to examine TRPV1 protein expression via Western blotting. Fibromyalgia pain mice had significantly increased TRPV1 levels as well as increased pPKA, pPI3K, and pPKC (Figure 4A, * *p* < 0.05, *n* = 6). All these effects were decreased by EPA administration (Figure 2A, # *p* < 0.05, *n* = 6). Mice fibromyalgia pain also increased pAkt and pmTOR expressions in the ACC. EPA treatment revealed an analgesic effect accompanied by attenuating the overexpressions of pAkt and pmTOR. Furthermore, cold stress-initiated mice fibromyalgia simultaneously potentiated pERK, pP38, and pJNK overexpressions (Figure 4A, * *p* < 0.05, *n* = 6), and EPA reversed this phenomenon (Figure 4B, # *p* < 0.05, *n* = 6). Lastly, the protein levels of pCREB and pNF-kB in the ACC displayed similar response trends (Figure 4B, * *p* < 0.05, *n* = 6). In addition, there were lower colocalizations of TRPV1 and pERK in the normal mouse ACC than in the fibromyalgia group. This increase was attenuated by EPA treatment (Figure 4C; green, red, or yellow colors; *n* = 3).

### 2.5. The Increased Levels of TRPV1 and Associated Molecules in the mPFC in Fibromyalgia Mice Decreased with EPA Treatment

The mouse mPFC is known to be involved in fibromyalgia pain through TRPV1 and the associated mediators [21]. Fibromyalgia mice showed higher protein levels of TRPV1 (Figure 5A, * *p* < 0.05, *n* = 6), which EPA administration effectively decreased (Figure 2A, # *p* < 0.05, *n* = 6). Downstream protein kinases, such as pPKA, pPI3K, and pPKC, were simultaneously increased after fibromyalgia pain induction, which were then abrogated by EPA treatment. We further examined the expressions of pAkt and pmTOR, which were augmented by cold stress-induced pain (Figure 5A, * *p* < 0.05, *n* = 6), and further reversed by EPA treatment (Figure 5B, # *p* < 0.05, *n* = 6). EPA treatment also attenuated the overexpressions of pERK, pP38, pJNK, pCREB, and pNF-kB. Lower TRPV1 and pERK colocalizations were observed in the normal mouse mPFC compared to the fibromyalgia group. This increase was attenuated by EPA treatment (Figure 5C; green, red, or yellow colors; *n* = 3).

### 2.6. Effects of EPA on Fibromyalgia-Related Pathways and Associated Molecules in Cerebellum Lobules 5–7

After the original approval of fibromyalgia pain according to the applicable mice behavior data, the mice cerebellum brain regions were sampled in order to determine the linked protein alteration in the mice cerebellum lobules 5–7. The outcome of fibromyalgia on the TRPV1 pathway was investigated, and the linked molecular foundations of EPA were then examined. The normal mouse group was taken as the ideal value and normalized to 100%. In total, cerebellum lobules 5–7 were determined. TRPV1 expression was significantly higher in the fibromyalgia group (Figure 6, Figure 7 and Figure 8A, * *p* < 0.05, *n* = 6), an increase attenuated by EPA treatment (Figure 6, Figure 7 and Figure 8A, # *p* < 0.05, *n* = 6). In addition, as expected, the participation of pPKA, pPI3K, pPKC, pAkt, and pmTOR, which are the major pathways of MAPK, were all increased in the fibromyalgia mice and further reversed by EPA treatment. Lastly, a similar tendency was also observed in pERK, pP38, pJNK, pCREB, and pNF-kB (Figure 6, Figure 7 and Figure 8B). Finally, we observed lower colocalizations of TRPV1 and pERK in the normal mouse cerebellum than in the fibromyalgia group. This increase was attenuated by EPA treatment (Figure 6, Figure 7 and Figure 8C; green, red, or yellow colors; *n* = 3).

### 2.7. Chemogenetics Inhibition of SSC Area Significantly Alleviated Mice Fibromyalgia Pain

Figure 9A displays a significant FM-induced mechanical hyperalgesia at day 5 after induction (Figure 7A, *n* = 6). Moreover, the chemogenetics blockage of the SSC region also significantly diminished mechanical hyperalgesia (Figure 7A, *n* = 6). Regarding thermal hyperalgesia, fibromyalgia mice showed substantial thermal hyperalgesia compared to the basal condition (Figure 7B, *n* = 6). Furthermore, thermal hyperalgesia was presented in mice with an SSC inhibition after chemogenetics manipulation (Figure 7B, *n* = 6). Furthermore, we next showed that CB1 was not changed after chemogenetics treatment. Considerably, pPKA, pPI3K, pPKC, pAkt, pmTOR, pERK, and pNFkB were reduced after chemogenetics manipulations of both the SSC and ACC regions (Figure 7C,D, *n* = 6).

## 3. Discussion

In the present study, we examined the effect and mechanisms of EPA on mice fibromyalgia pain. We used cold stress to successfully generate a mouse model of fibromyalgia pain. Mechanical and thermal hyperalgesia can be attenuated by EPA oral administration. Our data show increased expressions of TRPV1 and related kinases in the mice thalamus, SSC, ACC, mPFC, and cerebellum. The phenomena were diminished by EPA administration. Our data reveal the positive effects of EPA on a mouse fibromyalgia pain model and recommend the clinical use of EPA for fibromyalgia.

EPA, which is an omega-3 polyunsaturated fatty acid, is a vital nutrient that has heart-protection, anti-inflammation, neuroprotective, triglyceride lowering functions. Although EPA is used as a medicine or dietary supplement, its molecular target and molecular arrangement are still unclear. Matta et al., reported that EPA can evoke inward currents in TRPV1-expressing oocytes, suggesting EPA is a direct ligand of the TRPV1 channel. Thus, EPA significantly regulates the TRPV1-related signaling pathway indicating that dietary supplementation with EPA is beneficial for the treatment of pain [34]. Our results indicate that EPA has an analgesic effect on fibromyalgia pain through the regulation of TRPV1 and associated molecules.

Perna et al., suggest that DHA and EPA derivates, such as resolvins (RvD1, RvD2, and RvE1), are endogenous anti-inflammatory lipid derivatives with analgesic effects on somatic pain through TRPV1 modulation. In addition, RvD2 attenuated the capsaicin-induced Ca^2+^ influx of rectal submucosal neurons of patients with irritable bowel syndrome [35]. RvD1, RvD2, and RvE1 diminished histamine-induced TRPV1 activation in peripheral dorsal root ganglion neurons delivering an analgesic effect [35]. A recent article mentioned that the application of EPA had an analgesic effect on acetic acid-induced writhing, hot plate, and formalin tests. The anti-inflammatory effect of EPA was confirmed by a carrageenan test. EPA also decreased the levels of cytokines in the blood samples of the mice after induced inflammation by carrageenan [36]. Ji et al., showed that resolvins, including RvD1, RvD2, and RvE1, were endogenous lipid mediators of acute inflammation from the DHA and EPA. Resolvins are powerful anti-inflammatory and analgesic factors in several animal models. RvE1 and RvD1 significantly reduced inflammatory and postoperative pain. They stated that resolvins diminished pain via the regulation of TRP ion channels in spinal cord synaptic transmissions. They further indicated resolvins could offer novel therapeutic approaches for relieving pain [37]. The current study showed increased protein levels of TRPV1 and related factors in the ascending pain pathway, including the thalamus, SSC, ACC, and mPFC in fibromyalgia mice, which EPA administration could decrease. In addition, the CB1 receptor was attenuated in this fibromyalgia model suggesting its crucial role in EPA analgesia. These data evidence the possible role of TRPV1 as a novel target for treating fibromyalgia pain.

Silva et al., showed that fish oil significantly inhibited mechanical and thermal hyperalgesia via decreasing tumor necrosis factor levels in the spinal cord. In addition, fish oil increased the sciatic functional index, electrophysiological recordings, and the number of myelinated fibers in the sciatic nerve [38]. In addition, Veigas et al., found that mice fed with fish oil had alleviated thermal nociception accompanied by reduced acid-sensing ion channel 1a (ASIC1a), ASIC3, and TRPV1 expressions in the L3–L5 regions of the dorsal root ganglion [39]. Jo et al., reported that the extracellular administration of EPA created a rapid and concentration-dependent inhibition of the Na+ channel. They also showed that EPA attenuated the mRNA levels of the Na+ channel after the administration of EPA in cultured human bronchial smooth muscle cells [40]. Nakajima et al., indicated that EPA significantly initiated a rapid and concentration-dependent suppression of the Na+ channel. EPA reliably attenuated the levels of mRNA for SCN9A and SCN8A in rat prostate cancer cell lines [41]. Kang and Leaf showed that the addition of EPA prohibited the expected tachyarrhythmias. They suggested that EPA also modulated sodium and calcium channels to serve as an anticonvulsant in brain cells [42].

Moreover, in another study, EPA increased the proliferation of neural stem cells and the expression of 2-arachidonylglycerol. Pretreatment with cannabinoid 1 or cannabinoid 2 receptor antagonists can further diminish such activities [43]. RvE1 is a derivate of the EPA-specialized pro-resolving lipid mediator, which has anti-inflammatory and analgesic effects. Suzuki et al., reported that chronic pain was significantly induced after spared nerve injury and was further diminished by the intracerebroventricular injection of RvE1. The analgesic effect of RvE1 was observed through chemerin receptor ChemR23 [44]. Recent articles indicate that the intrathecal pre-treatment of RvE1 prohibits the induction of nerve injury-induced nociception and increases in Iba-1 (microglial marker) and TNF-α (brain inflammation) in mice spinal cords. The intrathecal injection of RvE1 at 3 weeks after neuropathic pain induction reliably abridged mechanical allodynia and heat hyperalgesia [45,46].

Bagher et al., reported that stimulating the CB1 receptor by orthosteric agonists had significant effects on alleviating the pain and neuroinflammation in chemotherapy-induced peripheral neuropathy (CIPN). They further verified that newly synthesized GAT229, a pure CB1-positive allosteric modulator alleviated neuropathic pain and slowed the progression of CIPN. GAT229 also abridged proinflammatory cytokines in mice dorsal root ganglia neurons through the brain-derived neurotrophic factor and nerve growth factor levels in mice DRG neurons [47]. Thapa et al., determined that painful responses were observed following the capsaicin stimulation of injured corneas in mice. Corneal neutrophil infiltration was also analyzed. CB1 allosteric ligands reliably alleviated pain responses to capsaicin activation. These nociceptive responses can be further reversed by CB1 antagonist AM251 suggesting the crucial target of the CB1 receptor [48]. Sierra et al., identified that the CB1 receptor and opioid ligands present noteworthy diminutions of pain behaviors in CIPN mice better than separate ligands. This tendency can be blocked by the CB1R-opioid antibody [49]. In the present study, we indicated that EPA administration significantly diminished mice fibromyalgia pain. Consistent with our hypothesis, the CB1 receptor decreased in specific regions in the mice with fibromyalgia pain. Furthermore, these effects can be reversed by EPA administration. Our results show that EPA has anti-nociceptive effects associated with CB1 signaling in a mouse brain. Thus, these targets can be potential treatments for fibromyalgia pain.

## 4. Materials and Methods

### 4.1. Animals and EPA Administration

We used 27 female C57BL/6 mice (8–12 weeks old) in this study (BioLASCO, Taipei, Taiwan; Taiwan Co., Ltd., Taipei, Taiwan). The mice underwent a 12 h light–dark cycle with food and water ad libitum. We used a statistic method to calculate the sample size. Nine mice in each group were considered as the minimum number required for a significance alpha level of 0.05 and a power of 80%. Mouse use was allowed by the Institute of Animal Care and Use Committee of China Medical University (Permit no. CMUIACUC-2023-071), Taiwan, following the Guide for the Use of Laboratory Animals (National Academy Press, Washington, DC, USA). Mice were randomly grouped into three groups: normal mice (Group 1: Normal), fibromyalgia mice (Group 2: FM), and fibromyalgia mice treated with EPA (Group 3: FM + EPA). The research was designed to euthanize the mice at the end of day 5. Nociceptive behaviors were evaluated at day 0, day 3, and day 5. EPA (300 mg/kg/day) was orally administered daily during the treatment period on days 3–5 [21].

### 4.2. Intermittent Cold Stress Model

All mice were kept at room temperature, 24 ± 1 °C, before the experiments. In the intermittent cold stress-induced fibromyalgia model, not in the normal Group, 2 mice were caged in a plexiglass cage (13 × 18.8 × 29.5 cm) covered with a stainless-steel mesh. On the first day (day 0), the mice were kept in a cold room at 4 °C overnight (4:00 p.m–10 a.m). The mice were next moved to a 24 °C room for 30 min at 10 a.m. After 30 min, the mice were then moved back to the cold room at 4 °C for 30 min. This process was repeated till 4:00 p.m. The mice were then placed in the 4 °C cold room overnight. Normal mice were kept at room temperature from days 0 to 5 of the experiment, with no Interventions.

### 4.3. Pain Behavior Test

Mechanical and thermal nociceptive behaviors were evaluated three times on day 0, day 3, and day 5 before and after fibromyalgia pain induction. All mice were placed in the behavior examination room for over 30 min to calm them down. Their behavior was analyzed only when the mice were quiet and were not sleeping or grooming. We first verified the von Frey filament examination; mechanical pain was tested by calculating the strength of responses to stimuli within 3 performances via the electronic von Frey test (IITC Life Science Inc., Los Angeles, CA, USA). Rodents were placed in a steel mesh (75 × 25 × 45 cm) and isolated with a plastic cage (10 × 6 × 11 cm). Rodents were confirmed by a filament examination of the central zone of the right back paw. The forces were calculated as grams and were documented when the mice withdrew their paws. Moreover, the Hargreaves test was performed to measure thermal nociceptive behavior by calculating the time to thermal stimuli with 3 presentations (IITC Life Sciences, SERIES8, Model 390G, CA, USA). The animals were placed in a cage on top of a glass area. All rodents were placed in the experimental room and were adapted to the condition over 30 min.

### 4.4. Western Blot Analysis

The whole thalamus, mPFC, SSC, ACC, and cerebellum tissues were excised to extract proteins. Tissues were placed on ice and stored at −80 °C until protein extraction. Total proteins were homogenized in cold radioimmunoprecipitation lysis buffer containing 50 mM of Tris-HCl pH 7.4, 250 mM of NaCl, 1% of NP-40, 5 mM of EDTA, 50 mM of NaF, 1 mM of Na3VO4, 0.02% of NaN3, and 1× protease inhibitor cocktail (AMRESCO). The extracted proteins were subjected to 8% SDS-Tris glycine gel electrophoresis and transferred to a polyvinylidene difluoride (PVDF) membrane. The membrane was blocked with 5% nonfat milk in Tris Buffered Saline with Tween (TBST) buffer (10 mM of Tris pH 7.5, 100 mM of NaCl, 0.1% Tween 20), incubated with a primary antibody in TBS-T with 1% bovine serum albumin (BSA) for 1 h at room temperature. Peroxidase-conjugated anti-rabbit antibody, anti-mouse antibody, or anti-goat antibody (1:5000) were used as appropriate secondary antibodies (Appendix A). The bands were visualized using an enhanced chemiluminescent substrate kit (PIERCE) with LAS-3000 Fujifilm (Fuji Photo Film Co., Ltd., Tokyo, Japan). Where applicable, the image intensities of specific bands were quantified using NIH ImageJ software 1.54h (Bethesda, Rockville, MD, USA). β-actin or α-tubulin were used as internal controls.

### 4.5. Immunofluorescence

The mice were euthanized using 5% isoflurane and intracardially perfused with normal saline followed by 4% paraformaldehyde. The brain was immediately dissected and postfixed with 4% paraformaldehyde at 4 °C for 3 days. The tissues were placed in 30% sucrose for cryoprotection overnight at 4 °C. The brain was embedded in an optimal cutting temperature compound and rapidly frozen using liquid nitrogen before storing the tissues at −80 °C. Frozen segments were cut into 20 mm sections using a cryostat and instantaneously placed on glass slides. The samples were fixed with 4% paraformaldehyde and incubated with a blocking solution, consisting of 3% BSA, 0.1% Triton X-100, and 0.02% sodium azide, for 1 h at room temperature. After blocking, the samples were incubated with the primary antibody TRPV1 (1:200, Alomone) and pERK (1:200, Millipore), prepared in a 1% BSA solution overnight. Then, the samples were incubated with the secondary antibody (1:500), Alexa Fluor™488-conjugated AffiniPure donkey anti-rabbit IgG (H + L), and Alexa Fluor™594-conjugated AffiniPure donkey anti-mouse IgG (H + L) (Appendix A) for 2 h at room temperature before being fixed with coverslips for immunofluorescence visualization.

### 4.6. Chemogenetic Operation

All mice were anesthetized with isoflurane and then fixed to a stereotaxic device. A 23G, 2 mm stainless cannula was inserted into the mice’s SSC regions, 0.5 mm posterior and 1.5 mm lateral to the bregma, 175 μm below the cortical surface, and making the skull immobile with dental cement. The injection cannula was inserted and connected to the Hamilton needle via a PE tube to inject 0.3 μL of viral solution for more than 3 min through a pump (KD Scientific, Holliston, MA, USA). After the injection, the injection cannula was well-maintained at the SSC for an additional 2 min to permit the solution to diffuse. The hM4Di DREADD (designer receptors exclusively activated by designer drugs: AAV8-hSyn-hM4D(Gi)-mCherry; Addgene Plasmid #50477) solution was injected into the SSC over two weeks. Clozapine N-oxide (CNO; Sigma C0832, St. Louis, MI, USA) was injected to stimulate the DREADD. CNO was thawed in 5% dimethyl sulfoxide (DMSO; Sigma D2650, St. Louis, MI, USA) and diluted with normal saline before the intraperitoneal injection of 1 mg/kg on day 3.

### 4.7. Statistical Analysis

Statistical analysis was performed using the SPSS statistic program. All statistic data are presented as the mean ± standard error (SEM). A Shapiro–Wilk test was performed to test data normality. Differences among all groups were tested using an Analysis of Variance (ANOVA) test followed by a post hoc Tukey’s test. *p* < 0.05 was considered statistically significant.

## 5. Conclusions

In this research, we evaluated the therapeutic effect of EPA on fibromyalgia pain and its actual cellular mechanisms in a mouse model. We used intermittent cold stress to initiate a mouse fibromyalgia pain model, here mechanical and thermal hyperalgesia were observed via von Frey and Hargraves tests. Mechanical and thermal pain were reversed by EPA administration. The protein levels of the TRPV1 signaling pathway were augmented in the ascending pain pathway, including the thalamus, medial prefrontal cortex, somatosensory cortex, anterior cingulate cortex, and cerebellum. Interestingly, the CB1 receptor was reduced in this fibromyalgia mouse model. EPA ingestion significantly diminished these phenomena in the aforementioned areas. The chemogenetics technique was performed to show the analgesic effects on the SSC and ACC circuits through the TRPV1 downstream pathway. Therefore, we present innovative evidence for the use of EPA in a mice fibromyalgia pain model and recommend the clinical use of EPA.

## Figures and Tables

**Figure 1 ijms-25-02901-f001:**
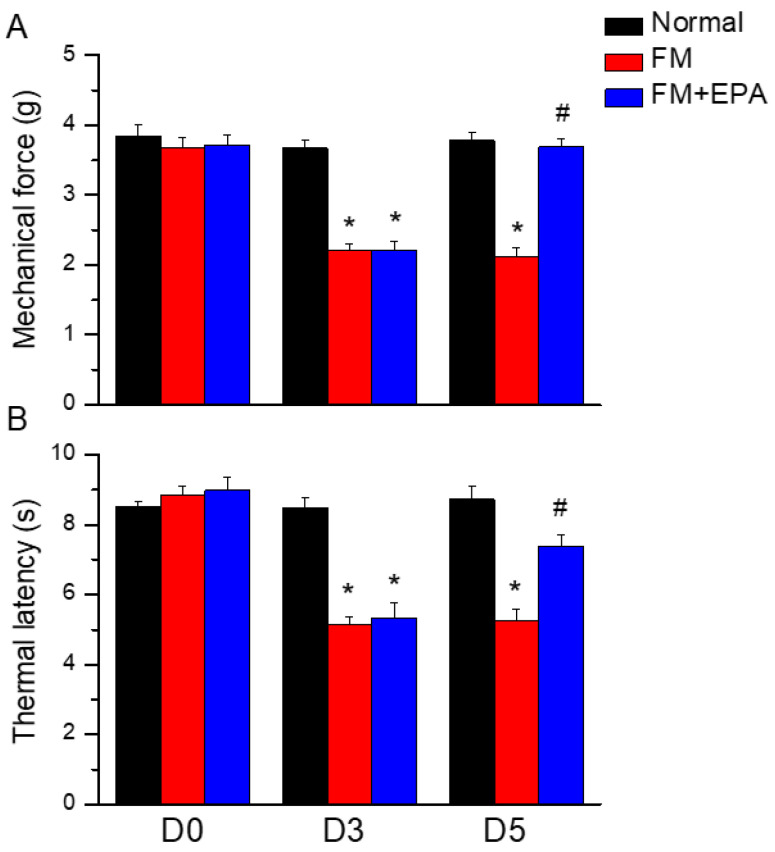
(**A**) Mechanical force measured with the von Frey filament test. (**B**) Thermal latency time via Hargreaves test. * *p* < 0.05 means significant difference in comparison to the normal group. ^#^ *p* < 0.05 means significant differences with the FM group. *n* = 9.

**Figure 2 ijms-25-02901-f002:**
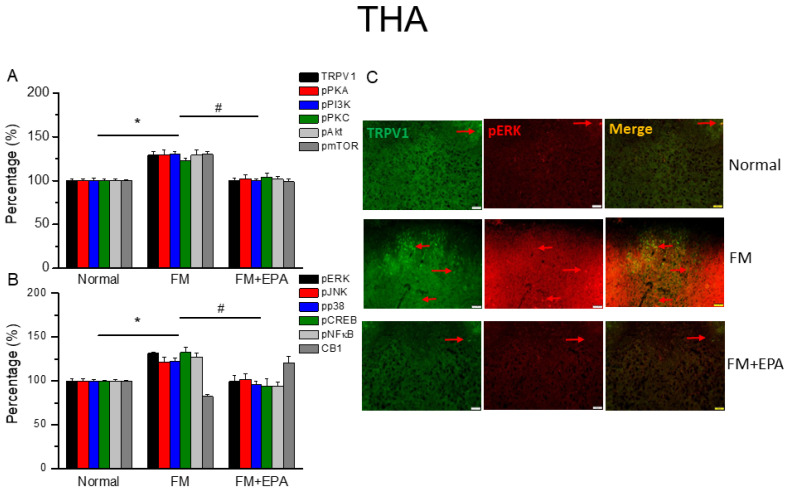
The expression levels of TRPV1 and related molecules in the mouse thalamus. Western blotting results of (**A**) TRPV1, pPKA, pPI3K, pPKC, pAkt, and pmTOR. (**B**) pERK, pJNK, pp38, pCREB, pNF-κB, and CB1 protein levels (refer to Appendix A for the Western blot bands). Red arrow means immune-positive signals. * *p* < 0.05 means significant difference in comparison to the normal group. ^#^ *p* < 0.05 means significant differences with the FM group. *n* = 9. (**C**) Immunofluorescence labeling of TRPV1, pERK, and double staining in the mouse THA (green, red, or yellow, respectively). Bar, 100 μm. *n* = 3 in all groups.

**Figure 3 ijms-25-02901-f003:**
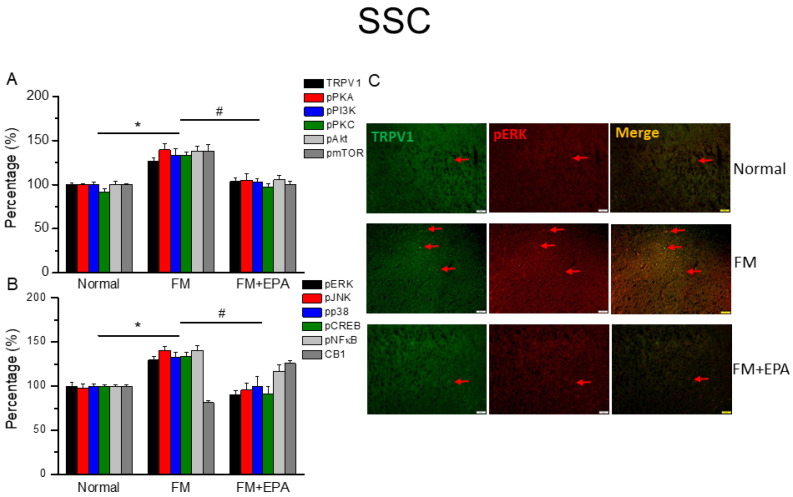
The expression levels of TRPV1 and related molecules in the mouse SSC. Western blotting results of (**A**) TRPV1, pPKA, pPI3K, pPKC, pAkt, and pmTOR. (**B**) pERK, pJNK, pp38, pCREB, pNF-κB, and CB1 protein levels (refer to Appendix A for the Western blot bands). Red arrow means immune-positive signals. * *p* < 0.05 means significant difference in comparison to the normal group. ^#^ *p* < 0.05 means significant differences with the FM group. *n* = 9. (**C**) Immunofluorescence labeling of TRPV1, pERK, and double staining in the mouse SSC (green, red, or yellow, respectively). Bar, 100 μm. *n* = 3 in all groups.

**Figure 4 ijms-25-02901-f004:**
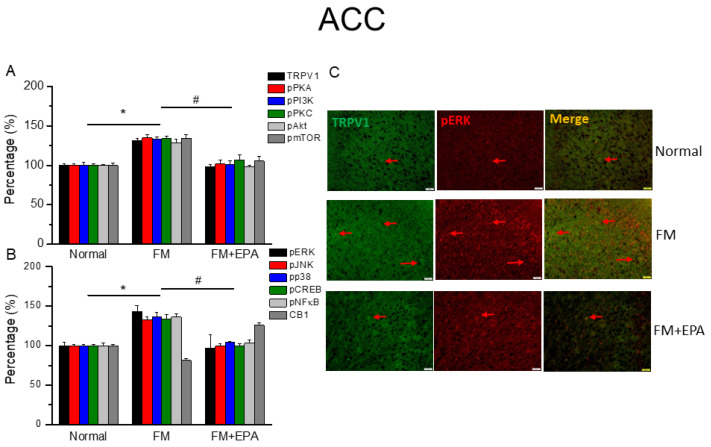
The expression levels of TRPV1 and related molecules in the mouse ACC. Western blotting results of (**A**) TRPV1, pPKA, pPI3K, pPKC, pAkt, and pmTOR. (**B**) pERK, pJNK, pp38, pCREB, pNF-κB, and CB1 protein levels (refer to Appendix A for the Western blot bands). Red arrow means immune-positive signals. * *p* < 0.05 means significant difference in comparison to the normal group. ^#^ *p* < 0.05 means significant differences with the FM group. *n* = 9. (**C**) Immunofluorescence labeling of TRPV1, pERK, and double staining in the mouse ACC (green, red, or yellow, respectively). Bar, 100 μm. *n* = 3 in all groups.

**Figure 5 ijms-25-02901-f005:**
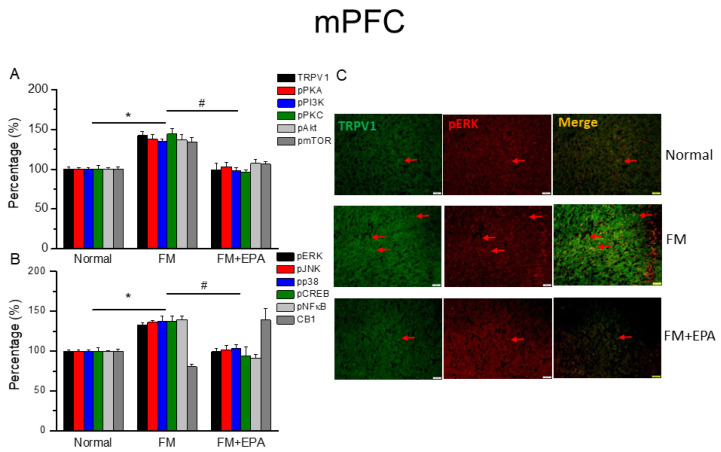
The expression levels of TRPV1 and related molecules in the mouse mPFC. Western blotting results of (**A**) TRPV1, pPKA, pPI3K, pPKC, pAkt, and pmTOR. (**B**) pERK, pJNK, pp38, pCREB, pNF-κB, and CB1 protein levels (refer to Appendix A for the Western blot bands). Red arrow means immune-positive signals. * *p* < 0.05 means significant difference in comparison to the normal group. ^#^ *p* < 0.05 means significant differences with the FM group. *n* = 9. (**C**) Immunofluorescence labeling of TRPV1, pERK, and double staining in the mouse mPFC (green, red, or yellow, respectively). Bar, 100 μm. *n* = 3 in all groups.

**Figure 6 ijms-25-02901-f006:**
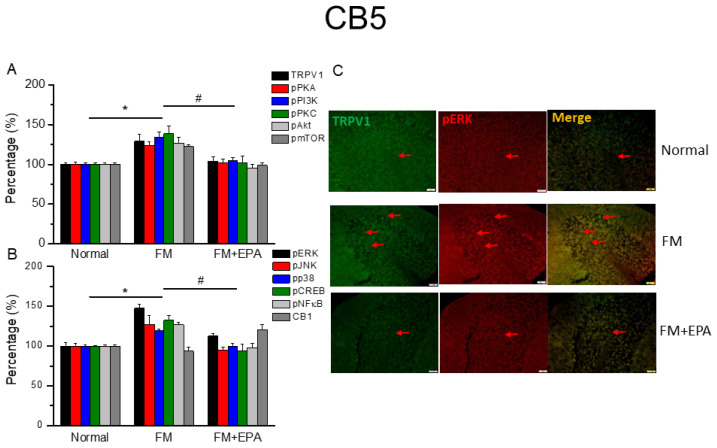
The expression levels of TRPV1 and related molecules in mouse cerebellum 5. Western blotting results of (**A**) TRPV1, pPKA, pPI3K, pPKC, pAkt, and pmTOR. (**B**) pERK, pJNK, pp38, pCREB, pNF-κB, and CB1 protein levels (refer to Appendix A for the Western blot bands). Red arrow means immune-positive signals. * *p* < 0.05 means significant difference in comparison to the normal group. ^#^ *p* < 0.05 means significant differences with the FM group. *n* = 9. (**C**) Immunofluorescence labeling of TRPV1, pERK, and double staining in mouse cerebellum 5 (green, red, or yellow, respectively). Bar, 100 μm. *n* = 3 in all groups.

**Figure 7 ijms-25-02901-f007:**
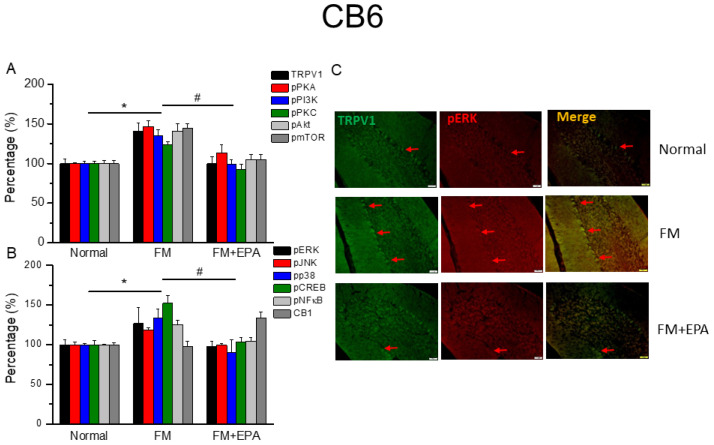
The expression levels of TRPV1 and related molecules in mouse cerebellum 6. Western blotting results of (**A**) TRPV1, pPKA, pPI3K, pPKC, pAkt, and pmTOR. (**B**) pERK, pJNK, pp38, pCREB, pNF-κB, and CB1 protein levels (refer to Appendix A for the Western blot bands). Red arrow means immune-positive signals. * *p* < 0.05 means significant difference in comparison to the normal group. ^#^ *p* < 0.05 means significant differences with the FM group. *n* = 9. (**C**) Immunofluorescence labeling of TRPV1, pERK, and double staining in mouse cerebellum 6 (green, red, or yellow, respectively). Bar, 100 μm. *n* = 3 in all groups.

**Figure 8 ijms-25-02901-f008:**
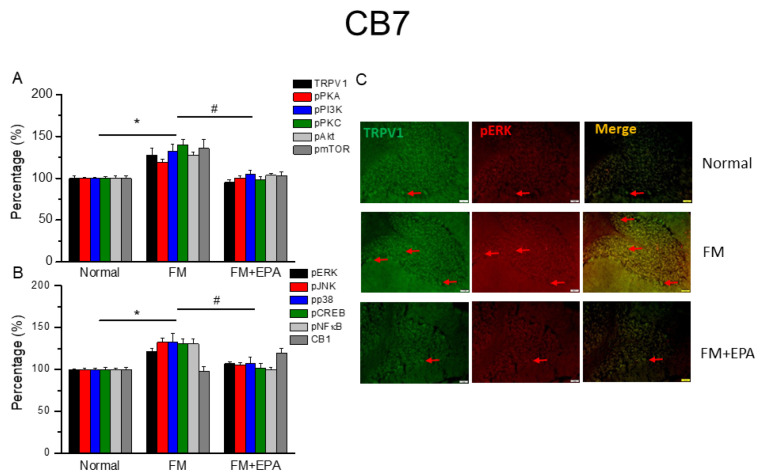
The expression levels of TRPV1 and related molecules in mouse cerebellum 7. Western blotting results of (**A**) TRPV1, pPKA, pPI3K, pPKC, pAkt, and pmTOR. (**B**) pERK, pJNK, pp38, pCREB, pNF-κB, and CB1 protein levels (refer to Appendix A for the Western blot bands). Red arrow means immune-positive signals. * *p* < 0.05 means significant difference in comparison to the normal group. ^#^ *p* < 0.05 means significant differences with the FM group. *n* = 9. (**C**) Immunofluorescence labeling of TRPV1, pERK, and double staining in mouse cerebellum 7 (green, red, or yellow, respectively). Bar, 100 μm. *n* = 3 in all groups.

**Figure 9 ijms-25-02901-f009:**
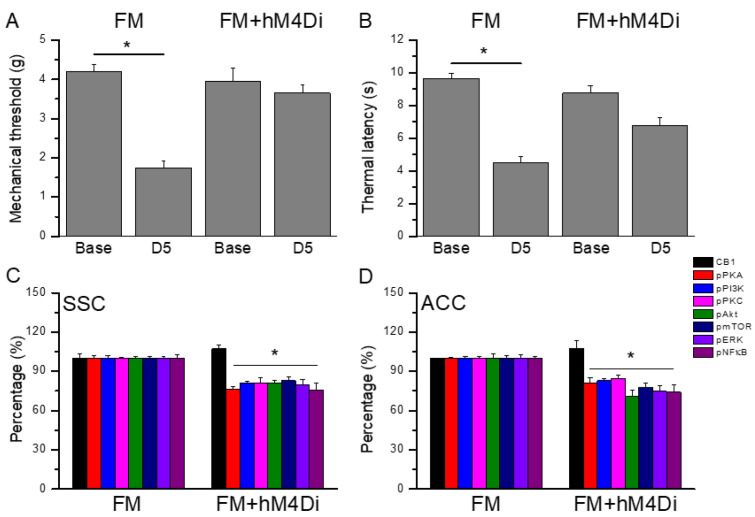
Pain behaviors of FM and FM mice treated with chemogenetics method (hM4Di). (**A**) Mechanical hyperalgesia (von Frey test). (**B**) Thermal hyperalgesia (Hargreaves test). * *p* < 0.05 means significant difference in comparison to basal condition. Protein percentages of CB1, pPKA, pPI3K, pPKC, pAkt, pmTOR, pERK, and pNFkB were measured in mice (**C**) SSCs and (**D**) ACCs. * *p* < 0.05 means significant difference in comparison to the FM group.

## Data Availability

The datasets supporting the conclusions of this article are included within the article.

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
