# Peer review of "Eicosapentaenoic Acid Modulates Transient Receptor Potential V1 Expression in Specific Brain Areas in a Mouse Fibromyalgia Pain Model"

_ijms, 2024, doi:10.3390/ijms25052901_

Round 1
Reviewer 1 Report
Comments and Suggestions for Authors
The article is devoted to studying the effect of Eicosapentaenoic acid on the expression of TRPV1 channels and a number of other proteins in the mouse brain in fibromyalgia pain model. The authors described the work done in sufficient detail and analyzed the results to show the effectiveness of using Eicosapentaenoic acid to reduce the effects of fibromyalgia. However, the article has a number of shortcomings.
The main criticisms include the incomplete section of research methods
1. There is no description of the intermittent cold stress model used in the methods section.
2. There is also no description of the chemogenetics method (hM4Di)
3. A set of all antibodies used and their combinations (primary-secondary) should be provided, indicating the manufacturer; alternatively, this can be given in a table in the sumplement.
3. Also in the discussion section, Eicosapentaenoic acid should be more fully characterized as a previously described direct ligand of the TRPV1 channel, which may explain the observed effects - for example, the article by Matta JA, Miyares RL, Ahern GP. TRPV1 is a novel target for omega-3 polyunsaturated fatty acids. J Physiol. 2007 Jan 15;578(Pt 2):397-411. doi: 10.1113/jphysiol.2006.121988
Minor remarks
1. In the abstract, it should be clarified that the given numerical values refer to the parameters of the von Frey test, or these values should be removed from this section.
2. line 63 – sentence broken by a dot
3. Line 64 – correct to TRPV1
4. there is no reference to Article 18 in the text
5. abbreviations should be deciphered when they first appear in the text - check for CB1 (line 98), IBa1 (line 157), THA should be in line 97, etc.
6. the same for proteins – line 74 – unify as further in the text (pERK, pp38 and so on)
7. line 256 – give a link to a known fact.
8. the figures should be reformatted so that the Western blots (bottom) and Immunofluorescence (right, not bottom) were correctly labeled (normal, FM, FM+EPK). Since Western blots are given in an additional file, there may be no need to duplicate them on a small scale in the main text, but rather present them in the form of a full-fledged supplement with appropriate captions for the figures.
9. Figure 9. C D “Protein intensities” or expression level?
Reviewer 2 Report
Comments and Suggestions for Authors
In this study, the investigators attempt to examine the potential modulation of eicosapentaenoic acid (EPA) on the protein expression of TRPV1 on certain brain areas in a mouse fibromyalgia pain model. EPA is a long-chain omega-3 polyunsaturated fatty acid found primarily in fish oil and certain algae, has been reported to alter the physical properties of cell membranes by incorporating into membrane phospholipids, as well as to modulate multiple type of transmembrane ion channels. Several queries regarding this work are shown below.
(1) Due to the abundance of evidence, eicosapentaenoic acid (EPA) has been shown to inhibit voltage-gated Na+ and Ca2+ currents in many cases, as well as suppress vasopressin-induced Ca2+ influx in aortic smooth myocytes (Asano et al., 1999:379; Rodrigo et al., J Mol Cell Cardiol 1999;31:733; Kang and Leaf, Am J Clin Nutr 2000;71(1 Suppl):202S; Jo et al., Biochem Biophys Res Commun 2005;331:1452; Nakajima et al., Br J Pharmacol 2009;156:420). Additionally, TRP channels also exhibit various types of non-selective ion channels. The activity of voltage-gated sodium channels has been strongly linked to nociceptive sensation (Jones et al., N Engl J Med 2023;389:393; O’Leary, Nat Med 2023; doi: 10.1038/d41591-023-00076-w). Therefore, the regulatory effect of EPA on the association of the expression level of TRPV1 channels with analgesic effects presented here might not be very specific. Alternatively, while EPA might inhibit vesicular nucleotide transporter.
(2) TRP channels are a diverse group of ion channels that play crucial roles in sensory signaling, including pain, temperature, taste, and vision. In the Introduction section of the manuscript. Please explain why the investigators chose the TRPV1 channel, why not other TRP channels such as TRPC, TRPV, TRPM, TRPA, and TRPP channels, while the text in lines 60-76 described the significance of TRPV1 channel. Additionally, TRPV channels are divided into TRPV1-TRPV5. Why is TRPV1 expression important for EPA treatment?
(3) EPA can alter the protein expression of TRPV1, but it does not mean that all these proteins are functional on the cell membrane; some might be non-functional. This point is also hoped to be understood by the investigators.
(4) The references shown above did not appear to be quoted in the manuscript.
(5) In lines 84, please replace “Noteworthy” with “Significant”. In lines 184-185, or the following figure legends, please add the P value for * and #.
(6) In line 146, please change “we” to “was”.
(7) In line 209 or 228, please change the subtitle of “Changing percentage …….”
(8) In lines 329-339, EPA-induced effects on vesicular nucleotide transporter (VNUT) could be overstated and the text needs to be removed, given that minimal experimental data have been shown in this manuscript.
(9) In line 340, “derivate” should be replaced with” derivatives”.
(10) In lines 340-342, the reference related to this sentence should be quoted, i.e., a paper of Perna et al.).
(11) In line 351, “meaningfully” could be inappropriately used.
(12) In lines 355-357, did the investigators doubt that EPA treatment can alter the expression of AS1Ca and ASIC3 channels?
(13) In lines 368-386, the stimulation of cannabinoid receptor 1 (CB1) has been raised. This issue appears to be irrelevant in the study. Do the investigators mean that EPA-induced modulation of TRPV1 is mediated by its binding to CB1 receptor? By using western blotting and immunofluorescence, TRPV1 expression was altered in several brain areas during the presence of EPA. The issue related to EPA binding to CB1 should be removed from the text of the manuscript. For instance, the text indicated in lines 348-349 needs to be removed. By the way, can the expression of TRPV1 caused by EPA treatment be reversed by pretreatment with CB1 antagonist AM251?
(14) The Discussion section of the manuscript needs to be re-organized, since the text appeared to be irrelevant to EPA treatment on either analgesic effect or TRPV1 expression.
Comments on the Quality of English LanguageMinor typographical errors need to be corrected.
Round 2
Reviewer 1 Report
Comments and Suggestions for Authors
The authors of the article took into account and corrected almost all the comments.
Apparently there is still some misunderstanding on the following issues:
1. all abbreviations in the text must be the same, correct abbreviations in line 71-72 from ERK, p38, JNK 9 to pERK, pp38, pJNK 9 – as used in following text
2. Links to support materials must be in the text of the article.
2a It is necessary to provide full captions to Western blot pictures in an additional file and refer to them in the text of the article
2b also in the methods section there should be a link to the table of antibodies used. The table itself should indicate complete information on all antibodies used, their names and catalog numbers; in addition, based on the text, “488-conjugated AffiniPure donkey anti-rabbit IgG (H+ L), 594-conjugated AffiniPure donkey anti-goat IgG (H + L), and peroxidase-conjugated AffiniPure donkey anti-mouse IgG (H + L)." were also used, they should also be added to the table
166-170 – no mention of pERK, the localization of which was also studied, correct it
